# STORM: An Integrated Framework for Fast Joint-Space Model-Predictive Control for Reactive Manipulation

**Mohak Bhardwaj[1,2], Balakumar Sundaralingam[1], Arsalan Mousavian[1], Nathan Ratliff[1],**

**Dieter Fox[1,2], Fabio Ramos[1,3], Byron Boots[1,2]**

[1]NVIDIA          [2]University of Washington          [3] University of Sydney

**Abstract:** Sampling-based model-predictive control (MPC) is a promising tool for feedback control of robots with complex, non-smooth dynamics, and cost functions. However, the computationally demanding nature of sampling-based MPC algorithms has been a key bottleneck in their application to high-dimensional robotic manipulation problems in the real world. Previous methods have addressed this issue by running MPC in the task space while relying on a low-level operational space controller for joint control. However, by not using the joint space of the robot in the MPC formulation, existing methods cannot directly account for non-task space related constraints such as avoiding joint limits, singular configurations, and link collisions. In this paper, we develop a system for fast, joint space sampling-based MPC for manipulators that is efficiently parallelized using GPUs. Our approach can handle task and joint space constraints while taking less than 8ms (125Hz) to compute the next control command. Further, our method can tightly integrate perception into the control problem by utilizing learned cost functions from raw sensor data. We validate our approach by deploying it on a Franka Panda robot for a variety of dynamic manipulation tasks. We study the effect of different cost formulations and MPC parameters on the synthesized behavior and provide key insights that pave the way for the application of sampling-based MPC for manipulators in a principled manner. We also provide highly optimized, open-source code to be used by the wider robot learning and control community. Videos of experiments can be found at: https://sites.google.com/view/manipulation-mpc

## 1 Introduction

Real-world robot manipulation can greatly benefit from real-time perception-driven feedback control [1, 2]. Consider an industrial robot tasked with stacking boxes from one pallet to another or a robot bartender moving drinks placed on a tray [3]. In both cases, the robot must ensure object stability via perception while respecting joint limits, avoiding singular configurations and collisions, and handling task constraints such as maintaining orientation during transfer. This leads to a complex control problem with competing objectives that is difficult to solve in real-time.

Classic approaches to solving these tasks rely on operational-space control (OSC) [4, 5, 6], where the different task costs are formulated in their respective spaces and then projected into the joint space (i.e., the control space of the robot) via a Jacobian map. OSC methods are inherently local as they only optimize for the next time step while ignoring future actions or states.

MPC based approaches attempt to find a locally-optimal policy over a finite horizon starting from the current state using a potentially imperfect dynamics model. An action from the policy is executed on the system and the optimization is performed again from the resulting next state which can overcome the effects of model-bias. MPC has been successfully applied on real robotic systems allowing them to rapidly react to changes in the environment [7, 8, 9, 5, 10]. Existing MPC methods that operate in the joint space of a manipulator are limited to gradient-based approaches [11, 12]

5th Conference on Robot Learning (CoRL 2021), London, UK.

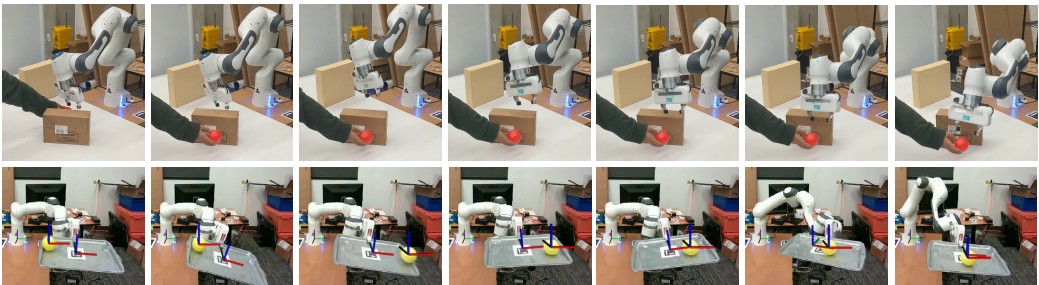

**Figure 1:** Our sampling-based model-predictive control framework operates in the joint space to enable a robot to achieve manipulation objectives such as tracking a moving target or balancing a ball on a plate while respecting constraints such as joint limits, singularity avoidance and collision avoidance via learned collision checking from raw sensor data.

which require the cost and dynamics to be differentiable. However, manipulation tasks often involve discontinuous phenomena such as contact and complex cost terms that are hard to differentiate analytically. Sampling-based methods such as Model-Predictive Path Integral (MPPI) [13] and Cross Entropy-Method (CEM) offer a promising alternative. Here, control sequences are sampled using a simple policy followed by rolling out the dynamics model to compute a sample-based gradient of the objective function. These algorithms make no restrictive assumptions about the cost, dynamics or policy class, are straightforward to parallelize and can be effectively applied on highly dynamic systems [10, 14]. These properties have also been a major factor in the increased adoption of sampling-based MPC by the Model-based RL community in recent years [15]

However, a key question still remains to be answered - *how well do these control algorithms transfer to high-dimensional robots like manipulators?* In this work, we describe an integrated system for sampling-based MPC that aims to answer this question. Our proposed framework, Stochastic Tensor Optimization for Robot Motion (STORM) implements a highly-parallelized control architecture that can optimize complex task objectives while simultaneously ensuring desirable properties such as smoothness, constraint satisfaction, and low control latency.

A major criticism of sampling-based MPC algorithms for full joint space control has been their inability to produce smooth (low-jerk) trajectories [16]. To address this challenge, we study different sub-components of sampling-based MPC and make several novel contributions such as low discrepancy action sampling, smooth interpolation and cost-function design, and demonstrate their effectiveness in scaling sampling-based methods to real robot manipulators. We also demonstrate that STORM can incorporate learned components in the control loop, by using learned self and environment collision costs. We integrate our system on a real Franka Panda robot arm where we demonstrate that feedback driven sampling-based MPC is able to solve complex and dynamic manipulation tasks with simple models. In summary, our major contributions are[1]

1. A novel sampling-based MPC with the introduction of low discrepancy sampling, smooth trajectory generation and behavior-based cost functions that are key for producing smooth, reactive, and precise robot motions.

2. A feedback control framework that directly integrates learned perception components into the control loop in the form of a learned self collision cost and a discrete collision checker between the robot links and raw environment pointcloud from [17].

3. An open-source and highly optimized implementation of sampling-based joint-space MPC, which achieves a control rate of 125Hz on a single GPU, a speedup of 100x compared to existing MPPI based manipulation implementations [17].

4. Empirical evaluation in simulation and a real-world Franka Panda robot on dynamic control tasks.

## 2   Problem Definition

We consider the problem of generating a feedback control law (or policy) for a robot manipulator with $d$ joints performing user-specified tasks in an unstructured environment where it is subject

---

[1]For further discussion of related work please refer to the supplementary material.

| Variable | Description |
|---|---|
| $\theta_t, \dot{\theta}_t, \ddot{\theta}_t$ | joint position, velocity, acceleration |
| $x_t = [\theta_t, \dot{\theta}_t, \ddot{\theta}_t]$ | robot state at time $t$ |
| $u_t$ | joint space command at time $t$ |
| $h \in [0, H)$ | number of steps in horizon $H$ |
| $n \in [0, N)$ | batch of control sequences |
| $K$ | iterations of optimization |
| $dt$ | change in time between steps in horizon |
| $\mathbf{dt}$ | vector of $dt$ $[0, H)$ |
| $\phi_t = \{\phi_{t,h}\}$ | parameters of policy at time t. |
| $\pi_{\phi_t}$ | distribution over control given state |
| $\mu_t = \mu_{t,h}$ | sequence of H Gaussian means |
| $\mathbf{\Sigma}_t = \Sigma_{t,h}$ | sequence of H Gaussian Covariances |
| $\mathbf{u} = \mathbf{u}_{n,h}$ | batch of N control sequences of length H |
| $\hat{\mathbf{x}} = \hat{\mathbf{x}}_{n,h}$ | batch of N state sequences of length H |
| $\hat{\mathbf{c}} = \hat{\mathbf{c}}_{n,h}$ | batch of N cost sequences of length H |
| $L$ | MPC loss function |

**Algorithm 1:** Sampling-Based MPC

**Input** : $\theta_0$,
**Parameter:** H, N, K

```
1  for t = 1...T do
2  |   x_t ← GET_STATE()
3  |   π_θ ← SHIFT()
4  |   for i = 1...K do
5  |   |   u ← SAMPLE_CONTROLS(π_{φ_t}, H, N)
6  |   |   x̂, ĉ, q̂ ← GENERATE_ROLLOUTS(x_t, H)
7  |   |   φ_t ← UPDATE_DISTRIBUTION(ĉ, u)
8  |   end
9  |   u_t = NEXT_COMMAND(π_φ)
10 |   EXECUTE_COMMAND(u_t)
11 end
```

**Figure 2:** We summarize the notations used on the left and the sampling based MPC algorithm on the right.

to non-linear constraints and must react in real-time to overcome errors due to its internal dynamics model, state estimation, and perception. The problem can be modelled as optimal control of a discrete-time stochastic dynamical system described by the equation, $x_{t+1} \sim P(x_{t+1}|x_t, u_t)$, where $P(x_{t+1}|x_t, u_t)$ defines the probability of transitioning into state $x_{t+1}$ conditioned on $x_t$ and control input $u_t \in \mathbb{R}^d$. The robot chooses controls using a deterministic or stochastic closed-loop policy $u_t \sim \pi(\cdot|x_t)$, incurs an instantaneous cost $c(x_t, u_t)$ and transitions into the next state, and the process continues for a finite horizon $T$. The goal is to design a policy that optimizes a task-specific objective function, $\pi^* = \arg\min_{\pi \in \Pi} \mathbb{E}_{\pi,P}\left[\sum_{t=0}^{T-1} c(x_t, u_t)\right]$ where $\Pi$ is the space of all policies.

The above setup is akin to optimizing a finite horizon Markov Decision Process (MDP) with continuous state and action spaces as done in reinforcement learning approaches [18]. Solving for a complex globally optimal policy is a hard problem especially since the task objective $c$ could be sparse or difficult to optimize. MPC can be viewed as a practically motivated strategy that simplifies the overall problem by focusing only on the states that the robot encounters *online* during execution and rapidly re-calculating a "simple" locally optimal policy. At state $x_t$, MPC performs a look-ahead for a shorter horizon $H < T$ using an approximate model $\hat{P}(\hat{x}_{t+1}|\hat{x}_t, a_t)$, approximate cost function $\hat{c}(x_t, a_t)$ and a parameterized policy $\pi_\phi$ to find the optimal parameters $\phi^*$ that minimize an objective function $L$

$$\phi^* = \arg\min_{\phi} L(\hat{c}, \hat{q}, \pi_\phi, \hat{P}, x_t) \tag{1}$$

where $\hat{q}(\cdot)$ is a terminal cost function that serves as a coarse approximation of the cost-to-go beyond the horizon. An action is sampled from $\pi_{\phi^*}$ and the optimization is performed again from the resulting state after applying the action to the robot. The optimization is hot-started from the solution at the previous timestep by using a *shift* operator, which allows MPC to produce good solutions with few iterations of optimization (usually 1 in practice).

In the next section, we first introduce a sampling-based MPC technique to solving the optimization in Eq. 1 and discuss the objective function, policy class, and update equations. We then present our approach for applying it to manipulation problems. An overview of the notation used in the paper is presented in Fig. 2.

## 3 Sampling-Based Model Predictive Control

Sampling-based MPC iteratively optimizes simple policy representations such as time-independant Gaussians over open-loop controls with parameters $\phi_t$ such that $\pi_{\phi_t} = \prod_{h=0}^{H-1} \pi_{\phi_{t,h}}$. Here, $\phi_t$ repre-

sent the sequence of means $\mu_t = [\mu_{t,0} \ldots \mu_{t,H-1}]$ and covariances $\boldsymbol{\Sigma}_t = [\Sigma_{t,0} \ldots \Sigma_{t,H-1}]$ at every step along the horizon $H$. A standard algorithm is shown in Fig. 2. At every iteration, the optimization proceeds by sampling a batch of $N$ control sequences of length $H$, $\mathbf{u}_{n \in [0,N), h \in [0,H)}$, from the current distribution (Line 5), followed by rolling out the approximate dynamics function using the sampled controls to get a batch of corresponding states $\hat{\mathbf{x}}_{n \in [0,N), h \in [0,H)}$ and costs $\hat{\mathbf{c}}_{n \in [0,N), h \in [0,H)}$ (Line 6). The policy parameters are then updated using a sample-based gradient of the objective function (Line 7). After $K \geq 1$ optimization iterations we can either sample an action from the resulting distribution or execute the first action from the mean (Line 9). We next describe how the distribution is updated. Consider the function $\hat{C}(\cdot)$,

$$\hat{C}(x_t, u_t) = \sum_{h=0}^{H-2} \gamma^h \hat{c}(\hat{x}_{t,h}, u_{t,h}) + \gamma^{H-1} \hat{q}(\hat{x}_{t,H-1}, a_{t,H-1}). \tag{2}$$

where $\gamma \in [0,1]$ is a discount factor that is used to favor immediate rewards. A widely used objective function is the exponentiated utility or the risk-seeking objective,

$$L = \mathbb{E}_{\pi_\theta, \hat{P}} \left[ \exp \left( \frac{-1}{\beta} \hat{C}(x_t, u_t) \right) \middle| \hat{x}_0 = x_t \right] \tag{3}$$

where $\beta$ is a temperature parameter. For this choice of objective, the mean and covariance are updated using a sample-based gradient as,

$$\mu_{t,h} = (1 - \alpha_\mu)\mu_{t-1,h} + \alpha_\mu \frac{\sum_{i=1}^N w_i u_{t,h}}{\sum_{i=1}^N w_i} \tag{4}$$

$$\Sigma_{t,h} = (1 - \alpha_\sigma)\Sigma_{t-1,h} + \alpha_\sigma \frac{\sum_{i=1}^N w_i (u_{t,h} - \mu_{t,h})(u_{t,h} - \mu_{t,h})^\top}{\sum_{i=1}^N w_i} \tag{5}$$

where $\alpha_\mu$ and $\alpha_\sigma$ are step-sizes that regularize the current solution to be close to the previous one and,

$$w_i = \exp \frac{-1}{\beta} \left( \sum_{h=0}^{H-2} \gamma^h \hat{c}(\hat{x}_{h,i}, a_{h,i}) + \gamma^{H-1} \hat{q}(\hat{x}_{H-1,i}, a_{H-1,i}) \right). \tag{6}$$

The update equation for the mean is the same as the well-known Model-Predictive Path Integral Control (MPPI) algorithm [13]. We refer the reader to [19] for the connection and derivation of update equations. While covariance adaptation has previously been explored in the context of Path Integral Policy Improvement [20] to automatically adjust exploration, standard implementations of MPPI on real systems generally do not update the covariance [13, 19]. However, we observed that updating the covariance leads to better performance with a fewer number of particles, such as stable behavior upon convergence to the goal. Once an action from the updated distribution is executed on the system, the mean and covariance are shifted forward with default values appended at the end to warmstart the optimization at the next timestep (Line 3).

The above formulation of MPC offers the flexibility to extract different behaviors from our algorithm by tuning different knobs such as the choice of approximate dynamics, running cost, terminal cost, the policy class and parameters such as the horizon length, number of particles, step sizes and discount factor $\gamma$. Next, we switch our focus to the domain of robot manipulation and build our approach for sampling-based MPC by systematically evaluating the effects of a subset of key design choices on the overall performance of the controller.

## 3.1 Approximate Model

The MPC paradigm allows us to effectively leverage simple models that are both computationally cheap to query and easy to optimize, as re-optimizing the control policy at every timestep can help to overcome the effects of errors in the approximate model. We leverage this error correcting property of MPC and utilize the kinematic model of the manipulator as our approximate transition model. Let the robot state be defined in the joint space as $x = [\theta, \dot{\theta}, \ddot{\theta}] \in \mathbb{R}^{3d}$ and the commanded action be the joint acceleration $u \in \mathbb{R}^d$. At every optimization iteration, we compute the state of the robot across the horizon ($\hat{\mathbf{x}} = [\Theta_{n,h}, \dot{\Theta}_{n,h}, \ddot{\Theta}_{n,h}]$, $h \in [0, H)$, $n \in [0, N)$) by integrating the sampled

control sequences $\mathbf{u}_{n\in[0,N),h\in[0,H)}$ from the robot's initial state $x_{\text{init}} = [\theta_{\text{init}}, \dot{\theta}_{\text{init}}, \ddot{\theta}_{\text{init}}]$ (i.e., current state of the real robot). This can be efficiently implemented as a tensor product followed by a sum:

$$\ddot{\Theta} = \mathbf{u} \qquad \dot{\Theta} = \dot{\theta}_{\text{init}} + S_l(1)\,\text{diag}(\mathbf{dt})\ddot{\Theta} \qquad \Theta = \theta_{\text{init}} + S_l(1)\,\text{diag}(\mathbf{dt})\dot{\Theta} \qquad (7)$$

where $S_l(1)$ is a lower triangular matrix filled with 1, and $\mathbf{dt}$ is a vector of delta timesteps across the horizon[2]. We use variable timesteps, with a smaller dt for the earlier steps along the horizon for higher resolution cost near the robot's current state and larger ones for later steps to get a better approximation of cost-to-go, similar to [8]. We intentionally write $\Theta_{n,h}$ as $\Theta$ to highlight the fact that we compute the states across the batch and horizon with a tensor operation without the need to iteratively compute the states across the horizon. This significantly speeds up the computation and is key to achieving the 8ms control latency.

Given a batch of states $\hat{\mathbf{x}}$, the Cartesian poses $\mathbf{X} \in \mathbf{SE}(3)$, velocities $\dot{\mathbf{X}}$, and accelerations $\ddot{\mathbf{X}}$ of the end-effector are obtained by using forward kinematics $FK(\Theta)$, the kinematic Jacobian $J(\Theta)$, and its derivative $\dot{J}(\Theta)$ as

$$X = FK(\Theta) \qquad \dot{X} = J(\Theta)\dot{\Theta} \qquad \ddot{X} = \dot{J}(\Theta)\dot{\Theta} + J(\Theta)\ddot{\Theta} \qquad (8)$$

We can compute the forward kinematics in batch as they depend only on the current state. This provides an easily parallelizable formulation of the robot model that is amenable to GPU acceleration.

## 3.2 Cost Function

The cost function $\hat{c}(s, a)$ encodes high-level robot behavior directly into the MPC optimization. This can be viewed as a form of cost-shaping that allows MPC to achieve sparse task objectives while also satisfying auxillary requirements such as avoiding joint limits, ensuring smooth motions and safety. We consider cost functions that are a weighted sum of simple cost terms, where each individual term encodes a desired robot behavior that can be easily implemented in a batched manner. We describe a subset of the cost terms here and refer the reader to the supplementary for more details.

### 3.2.1 Stopping for Contingencies

The finite horizon makes MPC myopic to events that can occur further in the future. Thus, it is desirable to ensure that the robot can safely stop within the horizon in reaction to events that might be observed at timestep $H-1$, especially in dynamic environments. We encode this behavior by computing a time varying velocity limit $\dot{\theta}_{max} \in \mathbb{R}^H$ for every timestep in the horizon based on a user-specifed maximum acceleration $\ddot{\theta}_{max}$ and the time until $H-1$. This means the joint velocity of the robot must allow it to come to a stop at the end of the horizon by applying the max acceleration. Any state that exceeds this velocity is penalized by a cost which is expressed as

$$\dot{\theta}_{max} = S_u(1)\ddot{\theta}_{max}dt \qquad \hat{c}_{\text{stop}}(\dot{\theta}_t) = \begin{cases} ||\dot{\theta}_{max,t} - |\dot{\theta}|||_2 & \text{if } \dot{\theta}_{max,t} - |\dot{\theta}| > 0.0 \\ 0, & \text{otherwise} \end{cases} \qquad (9)$$

where $S_u(1)$ is an upper triangular matrix filled with 1.

### 3.2.2 Avoiding Cartesian Local Minima

The manipulability score describes the ability of the end-effector to achieve any arbitrary velocity from a given joint configuration. It measures the volume of the ellipsoid formed by the kinematic Jacobian which collapses to zero at singular configurations. Thus, to encourage the robot to optimize control policies that avoid future kinematic singularities, we employ a cost term that penalizes small manipulability scores [21, 22]

$$\hat{c}_{\text{manip}}(\theta_t) = \begin{cases} 1.0 - \sqrt{J(\theta_t)J(\theta_t)^\top}, & \text{if}\sqrt{J(\theta_t)J(\theta_t)^\top} < k_m \\ 0.0, & \text{otherwise} \end{cases} \qquad (10)$$

---

[2]We use semi-implicit Euler integration which provides more stable solutions than explicit Euler due to it being symplectic. See https://gafferongames.com/post/integration_basics/

### 3.2.3 Self Collision Avoidance

Computing self-collision between the links of the robot for a large number of configurations can be computationally expensive [23, 17]. Hence, similar to previous approaches [23, 17], we train a neural network that predicts the closest distance [3] between the links of the robot given a joint configuration ($\theta$). A difference in our approach (termed jointNERF) is the use of positional encoding (i.e.,$[\sin(\theta), \cos(\theta)]$) that improves the accuracy of the distance prediction [24]. We compute a cost term as: $\hat{c}_{\text{self-coll}}(\theta_t) = \max(0, \text{jointNERF}(\theta_t))$

### 3.2.4 Environment Collision Avoidance

Safe operation in cluttered environments requires a tight coupling between perception and control for collision avoidance. Gradient-based approaches [25] generally rely on either known object shapes or pre-computed signed distance fields that provide gradient information for optimization. However, our sampling-based approach can handle discrete costs and as such we explore collision avoidance without using signed distances. Specifically, we use a learned collision checking function from Danielczuk *et al.* [17] that operates directly on raw pointcloud data and classifies if an robot link pointcloud $pc_l$ is in collision with the environment pointcloud $pc_{env}$ given the robot link's pose $X_l$.

$$\hat{c}_{\text{coll}}(pc_l, pc_{env}, X_l) = \begin{cases} 1, & \text{if } collision, \\ 0, & \text{otherwise.} \end{cases} \tag{11}$$

## 3.3 Sampling Strategy for Control Sequences

The method used for sampling controls from the Gaussian policy can have a great impact on the convergence of the optimization and can help embed different desirable behaviors such as ensuring smoothness. Pseudorandom sequences typically used in Monte Carlo integration exhibit an undesirable clustering of sampled points which results in empty regions. Whereas low-discrepancy sequences, where *low discrepancy* refers to the degree of deviation from perfect uniform sampling, alleviate this problem by defining deterministic samplers that correlate each point to avoid groupings [27]. Halton sequences [28] are a widely used form of *low discrepancy* number generators that attempt to improve the rate of convergence of Monte Carlo algorithms, and are reported to achieve superior performance in high-dimensional problems [29]. In particular, the Halton sequence uses the first $p_i, \ldots, p_d$ prime numbers to define a sequence, $\mathbf{w}_1, \mathbf{w}_2, \ldots$, for integers $i \geq 0$ and $b \geq 2$ where $\mathbf{w}_i = (\phi_{p_1}(i), \ldots, \phi_{p_d}(i))$ and $\phi_b(i) = \sum_{a=1}^{\infty} i_a b^{-a}$, with $i = \sum_{a=1}^{\infty} i_a b^{a-1}$ for $i_0, i_1, \cdots \in \{0, 1, \ldots, b-1\}$ [4]. We incorporate Halton sequences for sampling controls that can provide a better estimate of the objective function gradient. Controls from the Halton sequence are sampled once at the beginning and then transformed using the mean ($\mu_t$) and Covariance $\mathbf{\Sigma}_t$ of the current Gaussian policy.

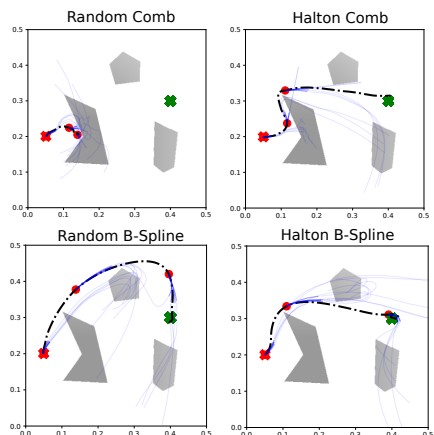

**Figure 3:** We compare our sampling scheme on a planar robot reacher task [26]. The holonomic robot must move from its initial position (green cross) to the desired position (red cross) while avoiding obstacles shown as the grey regions. The path taken by the robot is given by the black dot-dashed line and red circles are positions of the robot after every 30 timesteps. The blue lines are the rolled out trajectories of the top 5 particles. Our Halton B-Spline is able to find a smooth short path to the goal while Random B-Spline takes a longer path. Halton with a comb filter is not smooth as shown by the sudden path changes.

Furthermore, we explore two different strategies for enforcing smoothness in sampled control sequences. The first method is a *comb* filter that uses user-specified coefficients $[c_1, c_2, c_3]$ to filter out each sampled control trajectory along the horizon as $u_{t,h} = c_1 u_{t,h} + c_2 u_{t,h-1} + c_3 u_{t,h-1}$ This method has previously been used with sampling-based control techniques [30, 31], however, it requires extensive tuning and the filtered trajectories are not guaranteed to be smooth as neighboring samples in the horizon can have large difference in magnitude.

---

[3]Distance is positive when two links are penetrating and negative when not colliding.

[4]See http://extremelearning.com.au/unreasonable-effectiveness-of-quasirandom-sequences/ for a visualization of different low-discrepancy methods.

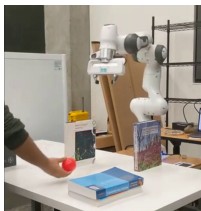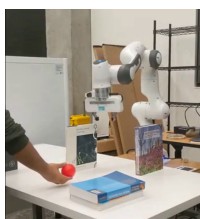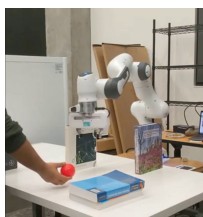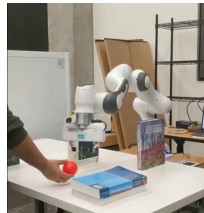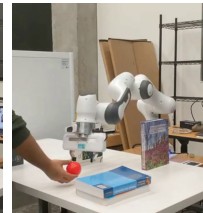

**Figure 4:** We show a sequence from our collision avoidance experiment where the robot has to move between thin walls to reach the orange ball held by the human. The robot tries to reach the ball but moves only as close as possible as any further motion would cause the elbow to hit the right book.

We propose an alternate strategy to enforce smoothness by fitting B-splines of degree 3 to controls sampled using a Halton Sequence. The resulting curve is sub-sampled at a finer resolution to obtain smooth joint acceleration samples which are then integrated to obtain corresponding joint velocity and position trajectories using Eq 7. In Fig. 3 we show a qualitative comparison between different combinations of sampling and smoothing strategies for a planar robot trying to reach a goal while avoiding obstacles where we see that our Halton + B-Spline sampling strategy is able to better explore the action space while maintaining smoothness (see supplementary for details).

**Covariance Parameterization**: Conventionally, sampling-based MPC algorithms such as MPPI parameterize the covariance of the Gaussian policy to be of the form $\Sigma_{t,h} = \sigma_u * I_{d\times d}$ where $\sigma_u$ is a scalar value and $I_{d\times d}$ is a $d \times d$ identity matrix, which forces the covariance to be the same across all control dimensions. However, in the case of manipulators it is desirable to allow the covariance of different joints to adapt independently so they can potentially optimize different cost terms such as pose reaching versus increasing manipulability. Thus we also consider covariance of the form $\Sigma_{t,h}^U = \sigma_u^T I_{d\times d}$ where $\sigma_u = [\sigma_1, \ldots, \sigma_d]$. Each term in $\sigma_u$ is then adapted based on the rollouts. Adapting the covariance along action dimensions has also been employed by [15] for CEM.

Our sampling strategy also offers us the flexibility of incorporating certain fixed set of action trajectories which could be task-dependent or even a library of pre-defined desired motions. We leverage this fact by incorporating a set of zero acceleration or "null" particles which allows the robot to coast at a constant velocity once the robot is accelerated sufficiently and also easily stop at the goal as demonstrated in our experiments.

# 4 Experimental Evaluation

Through our experiments we aim to analyze the effectiveness of STORM as a framework for real-time, perception-driven feedback control in real-world manipulation scenarios. To this end, we first study the performance of STORM in reacting to changing end-effector targets from perception data while satisfying task constraints such as maintaining user-specified orientation and avoiding obstacles in cluttered scenes. Second, we consider the dynamic task of balancing a ball on a tray grasped by the end effector that uses an approximate model of the ball dynamics. We also provide qualitative ablations in simulation for different components of our framework such as cost terms, sampling strategy and policy parameterization on our website (https://sites.google.com/view/manipulation-mpc). We additionally compare our approach to MOVEIT! and OSC [32] for the standard pose reaching problem in the supplementary material.

## 4.1 Tracking Moving Targets while Handling Task Constraints

A key strength of feedback-based MPC over "plan and execute" and OSC approaches is its ability to simultaneously optimize complex cost functions over a long horizon while demonstrating reactive behavior. We demonstrate this by having the robot react to changing goal poses obtained from noisy perception, while satisfying task constraints such as maintaining desired orientation and avoiding obstacles. In these experiments, a ball held by a human is tracked using a depth camera and the robot tries to reach the ball as the human moves it to different locations in the workspace.

**Obstacle Avoidance**: Demonstrating reactive motion and reasoning about obstacle avoidance in cluttered environments, while simultaneously coordinating large degrees of freedom leads to a hard online optimization problem. Standard planning and OSC approaches often fall short in optimizing for such behavior.

To test our method's capability to handle such scenarios, we setup two different table top environments, one consisting of two blocks representing common pick and place environments as shown in Fig. 1 and an environment with thin walls to represent a densely occupied space, as shown in Fig. 4. We use our perception based ball tracker to make the robot reach different positions in the environment. During the experiment, we also move the ball to some positions that are not reachable by the robot due to possibility of collision between the robot's links and the obstacles. As seen in Fig. 4, the robot handles these situations very well as it prioritizes collision avoidance over reaching the pose accurately. We present our full obstacle avoidance experiments as well as several experiments in simulation in the accompanying videos.

**Orientation Constraints**: Several common manipulation tasks such as moving a filled cup require maintaining orientation during motion which reduces the feasibility region of a controller. We test this scenario by imposing orientation constraints on the end-effector while tracking the ball. Our controller achieves a median quaternion error of $1.2485\%$ while tracking the ball with sufficient accuracy as shown in Fig. 5.

### 4.2 Dynamic Object Balancing

We consider a hard dynamic manipulation task where the robot tries to balance a ball placed on a tray grasped by a parallel jaw gripper as shown in Fig. 1. The location of the ball is tracked using the RGBD input from a RealSense camera at 30Hz. We use a simplified dynamics model of the object rolling on the tray under acceleration due to gravity and do not explicitly account for friction or inertial properties of the ball and do not perform any system identification. The purpose of this task is to demonstrate the robustness of MPC under severe model-bias while optimizing complex objectives and dealing with noisy perception. Fig. 6 shows the trajectory taken by the ball when the robot is controlled by STORM. MPC with its forward lookahead and rapid re-optimization is able to anticipate the ball's motion and correct for errors due to inaccurate dynamics and perception to center the ball on the tray with an error of less than 4cm.

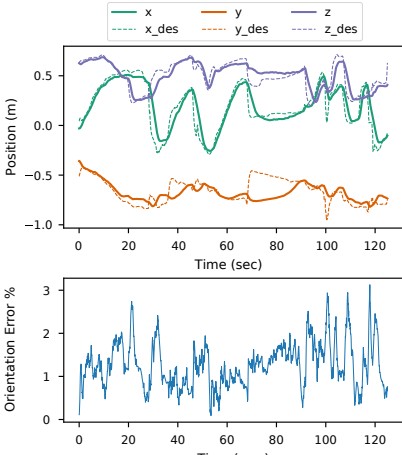

**Figure 5:** We move the ball across the workspace while having a high weight on maintaining a specific orientation of the end-effector. Our control scheme can maintain the orientation during motion as seen by the very low orientation error ($< 3\%$).

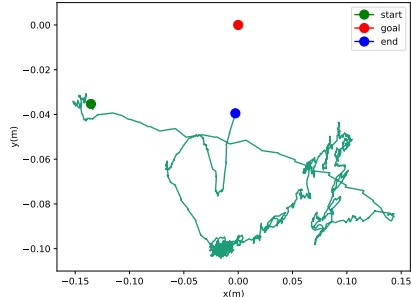

**Figure 6:** The observed trajectory of the ball in the frame of the tray for the balancing task is shown in the plot. Our controller is able to reach an error of less than 4cm using a simplified dynamics model and noisy perception inputs.

Our experimental results indicate that accurate and reactive manipulation behavior can be obtained by using relatively simple models and intuitive cost functions in an MPC framework. Sampling-based MPC also provides us the flexibility to encode different desired behaviors such as smooth motions directly in the optimization and tightly couple perception with control which are key components for real-world robot control.

### 5 Discussion

We presented a sampling-based MPC framework for manipulation that operates in the joint space and is able achieve smooth and reactive motions while respecting constraints, and demonstrated its performance on dynamic control tasks. The first key component of our approach is a fully tensorized kinematic model that allows for GPU-based acceleration of rollouts. Second, we leverage intuitive cost terms that encourage desirable behaviors. Third, our formulation allows us to leverage diverse sampling strategies to embed desirable properties directly into the optimization. However, a few key questions remain. First, performance can be made more robust by directly accounting for state uncertainty in the control loop. Second, at higher speeds, the kinematic model might induce significant model-bias. Here, learning a residual dynamics model [33] or a terminal Q-function [34] can help mitigate the effects of model-bias while still maintaining computational speed.

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
