# OpenReview forum: "STORM: An Integrated Framework for Fast Joint-Space Model-Predictive Control for Reactive Manipulation"
_robot-learning.org/CoRL/2021/Conference — CoRL2021 Oral_

### Official Review · Reviewer_U2Zx · 2021-07-07

**Originality:** Fair
**Technical Quality:** Good
**Clarity Of Presentation:** Good
**Impact:** 3

**Recommendation:**

Weak Accept: I recommend accepting the paper, but will not argue for my recommendation if the majority of other reviewers have a different opinion.

**Summary:**

The presented work introduces a set of elements to improve and fasten the performance of Sampling-Based MPC methods for high-dof robots. The proposed elements are: a novel sampling approach to guarantee smoothness action trajectories and good action space cover, integrating pretrained cost function and parallelizing the computation of the joint dynamics.

**Issues:**

The authors might focus on an ablation studies of the proposed elements, in particular it would interetsing an ablation study for the sampling methods. It is possible to define some qunatitative metrics to evaluate the performance of the Halton+B-Spline w.r.t. the others. Does it have an impact also in higher dimensional problems? Does it improve the jerkyness of the robot?

**Reviewer Expertise:**

Good: General knowledge of the area

**Strengths And Weaknesses:**

Strenghts:
1. The presented elements allows to boost the computational time of Sampling-based MPC methods 100 times.
2. Sampling Strategy: The paper proposes a novel sampling proceduce to guarantee smooth action trajectories and better action space cover while optimizing. The method seems interesting and they show qualitative results comparing the results sampling from normal distributions and using their proposed methods (Halton + B-Spline).

Weaknesses:
1. The paper lacks any reference to previous works trying to solve the problem of improving the computational efficiency of Samplign based MPC methods [1].
2. Approximate Model: The paper proposes to parallelize the integration of the trajectories. Given the initial robot state, a set of control actions, and given the discretized dynamics are linear with respect to the input, they parallelize the computation of the joint position and velocity trajectories. I do have two coments with respect to the approximate model: (i) It would be more accurate if the acceleration term also influence in the position, i.e. x(t) = x(0) + v(0)t + 0.5 a(0)t^2. and you can still parallelize the computation and (ii) This parallelization only works with linear systems. In case you want to model nonlinear dynamics, for example, the robot interaction with a certain underactuated environmental element, you require to evovle the dynamics sequentially. Also, I don't see any major novelty of parallelizing the evolution of linear dynamics.
3. Self Collision Avoidance: Computing self-collision might be computationally expensive. Thus, the paper proposes to use a pretrained neural network that predict the closest distance between the robot links. While it is a very interesting approach, this network has been previously presented in (in paper ref.[23])  and the paper focuses on applying it and adding minor modifications.
4. An important missing point in the paper is a proper set of metrics to evaluate the benefit of each integrated element in the overall framework. The paper could follow an ablation study to evaluate the benefit of the self-collision avoidance element or a quantitative metric to compare their sampling strategy w.r.t. gaussian policy. It would also interesting to provide the mathematical equation of the ball dynamics? Is it linear? Can you parallelize it in the optimization?

[1] Pinneri, Cristina, et al., "Sample-efficient Cross-Entropy Method for Real-time Planning", (2020)


**Summary Of Recommendation:**

The work is a set of individual elements to boost the computational time fro sample-based MPC methods. Evaluatiing each element individually, there is no a big contribution or novelty in any of them. The proposed approximate model  parallelization is standard and won't extend to nonlinear dynamics. Additionally, they should consider the acceleration effect in the joints position. The self-collision avoidance is just applying a previously presented work.

---

> ### Author Response · Authors · 2021-08-24
> **Response to review (1/2)**
>
> We thank the reviewer for taking the time to review our paper and providing thoughtful comments. However, we believe that the reviewer misunderstood the overall goals of the paper. The aim of the paper is to present an integrated control framework for applying sampling-based MPC to real world high-dof manipulation problems directly in the configuration space. We develop the first working instantiation of such a system, overcoming challenges due to computation and latency.  We firmly believe that such system implementations are crucial for advancing robotics research as they bring to light critical challenges that are often ignored by purely theoretical research. Our associated codebase also opens up avenues for followup learning-based research on real robots. Below, we address the concerns raised by the reviewer.
>
> **R4 W1: Reference to Prior Works**
>
> Thank you for bringing the paper that introduces iCEM to our notice. We will add this as a reference to the updated draft. However, although iCEM is able to significantly reduce the number of particles required for obtaining good performance, it still struggles to achieve real time control as shown in Table 2 of the paper. The implementation has a latency of 0.027, 0.163, 0.368 seconds per iteration for simulated HalfCheetah, Humanoid Standup and Fetch Pick and Place respectively with 100 trajectory samples and 32 CPU threads. On the contrary, our framework achieves a control latency of (0.008 seconds) 125Hz with 500 trajectory samples and works on a real world Franka Panda robot. A major reason for this 2x-36x times speed-up is that while [2] uses a (CPU-based) full dynamics simulator i.e, MuJoCo for trajectory rollouts, STORM is able to leverage a simplified kinematic model for fast compute on a GPU.
>
> Our kinematic model is able to provide sufficient accuracy to perform dynamic tasks on a real robot while optimizing complex cost functions constraints such as collision avoidance, joint limits etc. The simulated experiments in the iCEM paper use the perfect model hence it is unclear how their results would transfer to the real world.
>
> Furthermore, our proposed framework is flexible enough to incorporate the iCEM sampling strategy (mentioned by reviewer) and even use CEM as the underlying optimizer, if desired. In the current work, we selected MPPI as it has a history of strong performance on real robots [1] even though practitioners have been skeptical of their applicability to articulated manipulators. As Reviewer 1 also pointed out, we provide effective design choices to apply sampling based MPC to high DOF manipulators and incorporate learned components. Our paper fills a gap in current literature as, to the best of our knowledge, there does not exist any implementation of full joint-space sampling based MPC on real manipulators that accounts for different task spaces and constraints.
>
> **R4 W2: Nonlinear dynamics**
>
> There is always a trade-off in MPC between model accuracy and control latency. In the current work, we demonstrated that even with simple models, feedback control allows us to do collision free motion generation while optimizing complex cost functions. The choice of linear models is not a limiting factor; our framework also provides avenues to incorporate learned non-linear dynamics models that can further improve accuracy while maintaining computational efficiency. Specifically, we can overcome the need for sequential prediction of dynamics for non-linear systems by leveraging learning based techniques for predicting batched dynamics [3] as well as learning residual models over our base linear model such as [4].
>
> Thank you for pointing out that we can include acceleration into the position update for increased accuracy. We implemented the integration model you suggested and found no significant difference in the performance of the robot, mainly because our MPC runs at sufficiently fast rates to account for any errors in model approximation and our dt is small enough that the (0.5*a(t^2)) term only adds a small delta to the position.
>
> **R4 W3: Learned Self-Collision Avoidance**
>
> We agree that the specific network used for self-collision avoidance is not a new contribution, however, the integration of such a network in reactive model-predictive control on real manipulators is novel. Previously, such methods have only been demonstrated to work with IK solvers [5]. From a systems perspective, our framework is the first to demonstrate how learned self collision costs can be incorporated efficiently in the finite horizon planning cycle. Further, we are able to demonstrate how several complex objectives can be handled together while maintaining real-time control which is extremely hard with local optimization methods.

---

> > ### Author Response · Authors · 2021-08-25
> > **Response to review (2/2)**
> >
> > **R4 W4: Ablation Studies and Ball Dynamics**
> >
> > We have updated our website ( https://sites.google.com/view/manipulation-mpc/further-experimental-results ) with comprehensive quantitative ablation studies for different cost terms as well as MPPI parameters such as number of particles and the sampling strategy. We have also updated the supplementary material to provide the equations for the ball dynamics.
> >
> > **References**
> >
> > 1. Wagener, Nolan, Ching-An Cheng, Jacob Sacks, and Byron Boots. "An online learning approach to model predictive control." arXiv preprint arXiv:1902.08967 (2019).
> >
> > 2. Pinneri, Cristina, et al., "Sample-efficient Cross-Entropy Method for Real-time Planning", (2020)
> >
> > 3. Lambert, Nathan O., Albert Wilcox, Howard Zhang, Kristofer SJ Pister, and Roberto Calandra. "Learning Accurate Long-term Dynamics for Model-based Reinforcement Learning." arXiv preprint arXiv:2012.09156 (2020).
> >
> > 4. Saveriano, M., Yin, Y., Falco, P., & Lee, D. (2017, September). Data-efficient control policy search using residual dynamics learning. In 2017 IEEE/RSJ International Conference on Intelligent Robots and Systems (IROS) (pp. 4709-4715). IEEE.
> >
> > 5. Rakita, Daniel, Bilge Mutlu, and Michael Gleicher. "RelaxedIK: Real-time Synthesis of Accurate and Feasible Robot Arm Motion." In Robotics: Science and Systems, pp. 26-30. 2018.

---

> > ### Comment · Reviewer_U2Zx · 2021-09-02
> > **Response to the Authors**
> >
> > I thank the authors the answers but I am still not able to see the real contribution in the paper. You achieve high control frequencies in Sampling-based MPC, but with very minor addition.
> >
> > 1. Evolving Linear Dynamics in batch is trivial.
> > 2. The contribution of the design and modelling of a self collision network would be a contribution, but you are claiming contribution to using it as additional cost function in your objective.
> >
> > Could you explain  me in detail, where is the contribution in the points 1 or 2? I might be not properly understading the complexity of any of this two points.

---

> > > ### Author Response · Authors · 2021-09-02
> > > **Response to Reviewer**
> > >
> > > Thank you for your feedback. As stated in the paper and responses, this is a learning systems paper demonstrating how to achieve real-time stochastic MPC on a real robot arm. Our work brings together several threads of research into a single framework that produces impressive real-world results. This is non-trivial and far from obvious. Even the existing work on sample-efficient MPC (pointed out by the reviewer) does not achieve this level of scalability to dynamic real-world tasks. To the best of our knowledge, we are the first to provide such an implementation, which makes it novel.
> > >
> > >
> > > We agree with the reviewer that evolving linear dynamics in a batch **by itself** is trivial. However, this is typically not implemented in stochastic MPC methods. We argue for why this makes sense from a parallel computation perspective, demonstrate the impact and effectiveness of the approach on a real system, and, in the response above, present how it could be extended to nonlinear dynamics models if desired. The fact that we can perform complex dynamic tasks using simple linear dynamics is itself a strong result that also provides intuition for future research, i.e we should focus on using simple models combined with learning and parallel compute to efficiently trade-off computational speed with accuracy, which can make or break real-world robotic systems. The self-collision cost is indeed an existing tool that we leverage, however using it in the context of MPC is novel from a systems perspective, which is the primary focus of this paper. Furthermore, the dynamics model and self collision cost are only two components of our entire system, which integrates several different sub-components.
> > >
> > > Applications and systems are explicitly called out as contributions in the CoRL call for papers, and it could be argued that seminal works such as [1] do not make any novel theoretical contributions to a sub-field of robot learning or control, but are still important contributions to the field: they demonstrate how learning systems can bring together various algorithmic components to achieve novel results. We reiterate the need to encourage more such implementations to advance robotics research as it is often not clear how purely theoretical contributions (often evaluated in simulation) translate to the real-world. Thus, we believe that the novelty of the paper should be judged based on the overall framework, the quality of the real-world experimental evaluation, the usefulness of the design choices, and the future research directions that it opens up.
> > >
> > > [1] OpenAI - SOLVING RUBIK’S CUBE WITH A ROBOT HAND

---

> > > > ### Comment · Reviewer_U2Zx · 2021-09-02
> > > > **Response to the Authors**
> > > >
> > > > Thank for the fast answer.
> > > >
> > > > I agree that a systems paper would be enough contribution if the experiments were impressive. I feel quite unfair to compare the system's contribution of your work, with the OpenAI's rubik cube case. A good systems paper should bring strong experiments solving a complex task. OpenAI's rubik cube is solving a complex open problem as it is one of dexterous manipulation. Even if having real-time MPC is an important element, if your paper is required to be evaluated as a systems paper, I would expect you to bring a complex experiment with it. This is not the case. GoTo + Obstacle avoidance is a well-explored problem, that has been shown to be solvable by reactive motion generation [1] without the need for long-horizon planning. If the paper is considered a system paper, then, I consider the experiment is not well chosen, given long-horizon planning is not a must, to solve that specific problem.
> > > >
> > > > [1] Ratliff, N.D., Issac, J., Kappler, D., Birchfield, S. and Fox, D., 2018. Riemannian motion policies. arXiv preprint arXiv:1801.02854.

---

> > > > > ### Author Response · Authors · 2021-09-03
> > > > > **Response to Reviewer**
> > > > >
> > > > > Thank you for the feedback. Our aim was not to directly compare our method to OpenAI Rubik’s Cube as they both solve different problems, but to use it as a motivating example for the need to encourage strong systems implementations in robotics. We now address the points raised regarding experimental evaluation.
> > > > >
> > > > > First, it is disingenuous to characterize the experimental results entirely as GoTo + obstacle avoidance while completely ignoring the dynamic ball balancing task. The latter is a very difficult task to perform in the real world as it requires the controller to reason about the future consequences of actions and react in real time to errors due to model bias and noisy perception, all while simultaneously optimizing complex objectives in different task spaces. In order to satisfy these requirements, we need an MPC framework that can efficiently trade off computation time with accuracy. We demonstrate that our system can perform this task with multiple trials on a real robot arm without the need for extensive system identification for the ball. Please refer to Section 4.2 of our paper as well as Sec 4.1 of supplementary material. We also wish to highlight the disconnect with your evaluation of the experimental results compared with the other reviewers who found the experiments to be impressive, especially the ball balancing task, integration of learned components, and demonstration on a real robot.
> > > > >
> > > > > Second, the claim that reactive motion generation techniques such as Reimannian Motion Policies (RMPs) obviate the need for planning in pose reaching with obstacle avoidance is simply false. Although powerful, RMPs are inherently local (as they only perform 1-step optimization) and can therefore get stuck in local minima. In fact, the paper cited above [Ratliff et al., 2018] makes no claims that the local motion generation can solve long-range problems without either forward looking MPC-like optimizers or long-range heuristics. This is specifically discussed in Section V-C and Section V-D respectively. Furthermore, RMP-based methods require access to signed distance fields for collision avoidance which are computationally expensive to calculate. However, we demonstrate that our framework can use learned collision classifiers as cost functions, including ones that are inherently non-differentiable while maintaining high computation speed.
> > > > >
> > > > > We provide a qualitative comparison of RMPs and STORM reaching different goals in a cabinet on our website (https://sites.google.com/view/manipulation-mpc/further-experimental-results#h.ostrjfk36yjf). This example clearly demonstrates how RMPs can get stuck in local minima when the goal location is on the other side of the cabinet wall. On the contrary, MPC, owing to its forward lookahead, is able to escape such local minima and successfully reach the goal.

---

> > > > > > ### Comment · Reviewer_U2Zx · 2021-09-03
> > > > > > **Response to Authors**
> > > > > >
> > > > > > I thank the authors for their fast answers.
> > > > > >
> > > > > > Your answer has changed my mind and I am able to see the contribution of the work. I still do consider that the added modifications are not impressive (batch dynamics evolution and using pre-trained cost functions), but as a system paper, it is a good job and you have sufficient experiments to show the benefit of it.
> > > > > >
> > > > > > I will consider modifying my marks.
> > > > > >
> > > > > > Thanks for your patience and your fast and well-structured answers.

---

### Official Review · Reviewer_sfFw · 2021-07-22

**Originality:** Good
**Technical Quality:** Very Good
**Clarity Of Presentation:** Very Good
**Impact:** 3

**Recommendation:**

Strong Accept: I recommend accepting the paper and will argue for my recommendation even if other reviewers hold a different opinion.

**Summary:**

This paper proposes a MPC sampling-based controller that can be applied to real robotic manipulation tasks. The authors introduce several subcomponents to ensure that the tasks are solved using smooth and collision free movements. The system is tested both in simulation and on a robot, and the results look very promising.

**Issues:**

- In Section 3.2.3, why do the authors need a neural network for predicting the closest distance between the links of the robot? Can't this be computed using forward kinematics?
- In Figure 3, there seems to be an error in the caption where it says that initial and desired positions are shown in green and red, where it seems to be the other way around.
- It would be nice to see the effect of the learned components in the form of another ablation study.

Typos:
- Line 107: represent -> represents

**Reviewer Expertise:**

Good: General knowledge of the area

**Strengths And Weaknesses:**

Strengths:
- The approximate model used for evaluating action trajectories is straightforward and efficient enough for high-frequency robotic control.
- The idea of penalizing the trajectories that cannot stop at the end of horizon can be very effective for producing more energy-efficient motions.
- The paper is well-written and organized.

Weaknesses:
- The paper does not explain how the learned components for collision avoidance are trained.
- The paper claims that one advantage of their system is that it can incorporate learned components in the control loop. But to my understanding, that is the same for all MPC algorithms and I have seen a lot of works that use learned components to improve the performance/speed of their control systems.

**Summary Of Recommendation:**

Overall, I like this paper. The approach is interesting and the results look very promising. I'm interested in seeing how the learned components are helping. Maybe the authors can show this in a new ablation study.

---

> ### Author Response · Authors · 2021-08-25
> **Response to review**
>
> We thank the reviewer for carefully reviewing our paper and providing valuable suggestions for improvement. We address your concerns below.
>
> **R3 W1: Collision Avoidance Training**
>
> For the current experiments we used a pre-trained SceneCollisionnet model(https://github.com/NVlabs/SceneCollisionNet) provided by the authors of (https://arxiv.org/abs/2011.10726). The model was trained for table-top environments which is appropriate for our setting as well. We will clarify this in the updated draft after the reviewer discussion period.
>
> **R3W2: Learned Components in MPC**
>
> We agree that prior MPC approaches have demonstrated that learned components can be incorporated into the optimization. However, in this work we explore how we can bring the benefits of sampling based MPC to dynamic real-world manipulation tasks which pose critical systems and safety-related challenges. To this end, we provide a novel framework that uses simple known models in conjunction with learned cost functions for reactive motion generation in the full configuration space of the robot. We specifically wanted to explore if sampling based MPC can optimize complex cost terms (which can be learned classifiers) in various task spaces as this is necessary for reactive motion generation with manipulators. To study this, we chose to leverage the known kinematics of a robot with fast MPC for overcoming model errors as we do not yet have a way to learn accurate global dynamics models for manipulators that also can run fast enough for use in MPC. To the best of our knowledge, our work is the first to demonstrate that sampling based MPC can indeed handle non-smooth (classifier) and complex cost terms in many different task spaces and generate reactive motion in the joint space for application to real-world manipulators.
>
> **R3I1: Using a Learned Self Collision Cost**
>
> While collision can be computed using forward kinematics, this process is computationally slow in practice as we need to check collision between all pairs of links. This becomes a major computational bottleneck when we have to compute across many rollouts. We performed a timing benchmark for an increasing batch size of query configurations where the learned function is over 40 times faster on average with a low latency of 0.4-0.6ms even for large batch sizes of 20-40k. We provide these results on our website (https://sites.google.com/view/manipulation-mpc/further-experimental-results#h.3v3jt4qbc7zu).
>
> **R3I3: Ablation Studies**
>
> We have updated our project website to add ablation studies that demonstrate the effectiveness of learned components as well as different cost terms and MPC parameters on the controller performance.
> (https://sites.google.com/view/manipulation-mpc/further-experimental-results)

---

> > ### Comment · Reviewer_sfFw · 2021-09-03
> > **Thanks for your answers**
> >
> > Hi,
> >
> > I thank the authors for updating the paper and taking the reviews into account. I change my vote to strong accept.

---

### Official Review · Reviewer_poXv · 2021-07-23

**Originality:** Very Good
**Technical Quality:** Very Good
**Clarity Of Presentation:** Very Good
**Impact:** 4

**Recommendation:**

Strong Accept: I recommend accepting the paper and will argue for my recommendation even if other reviewers hold a different opinion.

**Summary:**

This paper proposes an integrated system for sampling based MPC for fast reactive manipulation control. Authors proposed different cost terms to achieve better manipulation behavior, also increase the sampling efficiency of sample-based MPC. The approach is validated both in simulation and on a real robot, and also authors plan to open source thier codebase.

**Issues:**

- Videos are too lengthy, could be shorter with more caption to describe what happens (or increase the playback speed)
- The last paragraph of section 3 says the sampling can also sample from a fixed set of predefined action trajectories, but it is not clear to me which experiments that explicitly demonstrate this point.
- Does the ball dynamics consider the collision between the edge of the tray and the ball? While the result is impressive, I found the statement of ball didn’t fall in all ten episodes not quite accurate. From the supplementary video, the ball did not fall off the tray clearly due to the edge of the tray tray to stop the ball, and it doesn’t seem that  the forward model takes such a contact model into account.
- In the supplementary, I can only see the computation time compared against OSC. Are there any performance comparisons? It would be good to show two video clips to demonstrate the difference between STORM and OSC formulation. And if you introduce potential field terms in nullspace projection, can OSC also perform a reasonable obstacle avoidance?


**Reviewer Expertise:**

Very good: Comprehensive knowledge of the area

**Strengths And Weaknesses:**

Strengths:
- A clear description of the system description
- A formulation to enable parallel computation of MPC on GPU, which make this MPC formulation work in real-time and on the real hardware.
- A sample efficient approach to consider a longer horizon motion than OSC, which benefits behaviors like avoiding obstacles.

Weakeness:
- As this is a sampling-based approach, it would have been better if authors can provide a quantitative comparison by varying the number of samples needed and the length of horizon. That would give readers a better idea how the approach performs in terms of the motion quality and the computation time by varying different essential parameters.
- For the cost term in 3.2.3 and 3.2.4, both cost terms rely on the neural network based functions. How the uncertainty introduced from neural network computation affects the algorithm? The experiments show the extrinsic evaluation of the cost terms, but it would be good to understand more intrinsically about the cost terms. Will there be any catastrophic collision due to incorrect computation from neural networks? If not, can it be guaranteed for such safety-critical terms.




**Summary Of Recommendation:**

In spite of some detailed concerns, I think this paper could have a significant contribution to the community, especially as a basic component for robot manipulation control.

---

> ### Author Response · Authors · 2021-08-24
> **Response to review**
>
> Thank you for taking the time to review our paper and providing valuable insights. We are glad that the reviewer found our work to be a significant contribution to the robot controls community. We address some of the concerns below
>
> **R2 W1: Ablations**
>
> We have updated our website to provide ablation studies of different components of our system. These experiments aim to provide insights to readers and practitioners on the impact of different parameters, cost functions and sampling strategies on the controller’s performance.
>
> **R2 W2: Safety guarantees using neural network costs**
>
> Our learned environment collision checking outputs a probability (0-1) of collision. Currently, we take uncertainty into consideration by thresholding at 0.4 instead of 0.5 to be extra cautious. We don’t have explicit measurements of uncertainty from the network. However, if we do have access to it, we can directly use it to weigh the collision cost. In regards to safety guarantees, there is work~(http://www.roboticsproceedings.org/rss14/p42.html) that leverages tube-MPC to provide behavior guarantees to MPPI which can be integrated in our framework.
>
>
> **R2 I2: Sampling fixed trajectories**
>
> Our experiments (both simulated and real-world) use zero acceleration or “null” particles as a fraction of total particles that enable the robot to stay at the goal without oscillations. In the future, we aim to incorporate other fixed control trajectories in the form of motion primitives into the sampling as well.
>
> **R2 I3:  Ball Model**
>
> The forward model indeed does not take the edge of the tray into account. However, the ball can still fall if the robot moves the tray too aggressively or does not correct for the ball pose based on perception feedback. In our experiments, we found that the ball does not go close to the edge too often unless that start position is near the edge (we have updated Figure 3 in supplementary to show the approximate boundary of the tray). However, the reviewer brings up a good point and in the future we will augment the ball model to incorporate the edge of the tray.
>
> **R2 I4: Comparison to OSC**
>
> We compared our approach to MMC [1]  which is a sophisticated operational space control method for pose reaching as shown in Table 2 in the supplementary material. In our real robot experiments, MMC repeatedly entered a self collision state when attempting to reach one of the poses. Although we can potentially incorporate self and environment collision avoidance with OSC, this usually requires extensive tuning between the different forces and computing signed distance fields (SDFs) which can be computationally expensive. Further, OSC methods are inherently local as they optimize for a single step action, which makes them susceptible to local minima such as causing oscillatory behavior when moving between two obstacles (https://ieeexplore.ieee.org/document/131810).
>
> **References**
>
> [1] Haviland, Jesse, and Peter Corke. "A purely-reactive manipulability-maximising motion controller." arXiv preprint arXiv:2002.11901 (2020).

---

> > ### Comment · Reviewer_poXv · 2021-09-03
> > **Final comment**
> >
> > I would like to thank the authors for the response. All points look reasonable to me, and I'll keep my score.

---

### Official Review · Reviewer_xm2M · 2021-07-23

**Originality:** Very Good
**Technical Quality:** Excellent
**Clarity Of Presentation:** Excellent
**Impact:** 4

**Recommendation:**

Strong Accept: I recommend accepting the paper and will argue for my recommendation even if other reviewers hold a different opinion.

**Summary:**

The paper provides a new framework for sampling-based model predictive control applied to robot manipulation. Contrary to previous MPC formulations, the authors employ MPC directly in the joint space, which allows them to explicitly incorporate collision or joint limit constraints. Planning is formulated as an MDP with continuous state and actions. To efficiently find a policy over a fixed time horizon in realtime, the authors use a simplified robot model by using forward dynamics and to leverage low-discrepancy sampling to find valid control sequences (policies) over the desired time horizon. Using several manipulation-related cost function, the authors demonstrate that their framework can be used to quickly plan joint paths for a fixed-base manipulator robot to reach dynamic target locations while avoiding obstacles. They also show that their framework can be applied to dynamical balancing tasks, where a ball is balanced on a tray.

**Issues:**

-- Please explain how you would make this method more rigorous in the future, i.e. what kind of guarantees might you be able to give if a practitioner wants to use this method.

-- Consider talking more about limitations of the approach and how we can overcome them in the future. For example, you never showed that the model-based approximation performs worse at higher speeds.

**Reviewer Expertise:**

Good: General knowledge of the area

**Strengths And Weaknesses:**

Strengths

-- The paper is well written and easily digestable. The authors did a fantastic job to motivate the problem, I really enjoyed reading the problem definition, which was easily understandable even though I am not an expert on reinforcement learning. The method section is self explanatory and I was able to follow their line of reasoning. While some details remained missing or had to be put in the supplementary material, I think the paper itself provides all the informations necessary to understand the ideas.

-- The experimental evaluation is great. While the applications are challenging and useful, the authors even made the extra step of providing real-world experiments on a real robot system. The balancing skills of the ball on the tray is impressive and it is a good showcase for their method. The videos nicely supplement and demonstrate the usefulness of their approach.

-- Reproducibility is often an issue with papers. The authors tackled this issue by providing code in the supplementary material, which would make it easy for other researchers to reproduce their results.

Weaknesses

-- It is clear to me that the authors have a deep understanding of what design choices actually work, in order to make the algorithm actually applicable. However, it would be great if you could (maybe in a later publication) try to provide more explanations of why
your specific design choices work.

-- So far the algorithm has been demonstrated on a fixed-base manipulator. It would be interesting to see if you can scale this approach to even more complex systems, for example mobile multi-robot or other robotic systems.

-- While the results are great, I would have liked to see more reactive scenarios, i.e. where you add obstacles on-the-fly, push the robot away or remove obstacles.

-- Similar to having more reactive scenarios, I would have liked to see more limitations of the approach, i.e. explicit scenarios where it fails.
I guess one example could be where the time horizon is too small and the robot takes a wrong path. It would be nice for other researcher to see how far this idea can be pushed and at what points new methods/ideas are needed.


**Summary Of Recommendation:**

The paper provides a novel method to achieve reactive manipulation using sampling-based MPC. The experiments are convincing and I think the whole approach holds a lot of potential for a broader range of applications. While I would like to see more reactive and limiting experiments, I think the paper itself is well-written, technically sound and should be a strong accept.

---

> ### Author Response · Authors · 2021-08-24
> **Response to review**
>
> Thank you for reviewing our paper and providing valuable feedback. We are glad that you found the paper well motivated and appreciated our real world experimental results. We address your concerns below:
>
> **R1 W1: Design Choices**
>
> We have run extensive ablations on the various design choices in our approach and have made it available on our website (https://sites.google.com/view/manipulation-mpc/further-experimental-results). We specifically ran ablations on the different weight scales for each cost term to see how it affects the performance and also ran ablations on the number of particles used for rollouts.
>
>
> **R1 W2: More Complex Robots**
>
> Our approach can scale to mobile manipulators directly as we only need to add a moving base to our forward model. However, we currently do not have access to a real mobile manipulator to run experiments.
>
> **R1 W3: Dynamic Obstacle Avoidance**
>
> We provide a qualitative demonstration of the system avoiding moving obstacles while reaching an end-effector target pose as a video on our website(https://sites.google.com/view/manipulation-mpc/further-experimental-results#h.1ftgxyyxsj65). Here, the manipulator rapidly reacts to avoid collision as the obstacle moves close to it and is able to leverage redundancy to optimize multiple task space costs.
>
>
> **R1 W4 & R1 I2: Limitations**
>
> As pointed out by the reviewer, one of the major limitations of MPC approaches can be the finite horizon which can cause it to get stuck in local minima. We demonstrate this qualitatively in this video (https://sites.google.com/view/manipulation-mpc/further-experimental-results#h.ugmxgp7scqxs). A possible solution to this is learning terminal Q-functions that can increase the effective horizon while mitigating model-bias.
>
> **R1 I1: Future Work**
>
> We plan to extend the framework along the following lines:
>
> 1. Overcoming model-bias and finite-horizon errors
>
> Several recent approaches have aimed at combining MPC with Reinforcement Learning (RL) by learning either dynamics models[1] or terminal value functions [2] or both [3]. Here MPC serves as an efficient exploration policy for RL and RL helps overcome model and finite horizon bias in MPC. These approaches can be readily incorporated in our framework to improve pose reaching accuracy as well as perform even harder dynamic tasks such as moving the tray to different poses while balancing the ball.
>
> 2. Uncertainty-aware planning
>
> One of the major issues we faced in our hardware implementation was noise in the state estimates (especially joint velocity) from the Franka Panda robot. This can cause a feedback controller to react to unwanted noise and deteriorate pose reaching accuracy. Intelligently incorporating state uncertainty into the planning cycle can help mitigate these issues.
>
> 3. Complex sampling distributions
>
> Our current MPPI controller uses a simple open-loop Gaussian as the control distribution. We are working on adding multi-modal distributions such as Gaussian Mixture Models and SV-MPC[4] which can provide better exploration.
>
> 4. Guarantees for Practitioners:
>
> We provide an empirical analysis of self collision and joint limit avoidance costs which shows that they can successfully prevent the robot from violating constraints (https://sites.google.com/view/manipulation-mpc/further-experimental-results#h.ud1wkx4676yl). In the future, we aim to perform a similar extensive analysis of the environment collision cost under a diverse range of scenarios.
>
>
> **References**
>
> 1. Lowrey, Kendall, Aravind Rajeswaran, Sham Kakade, Emanuel Todorov, and Igor Mordatch. "Plan Online, Learn Offline: Efficient Learning and Exploration via Model-Based Control." In International Conference on Learning Representations. 2018.
>
> 2. Chua, Kurtland, Roberto Calandra, Rowan McAllister, and Sergey Levine. "Deep Reinforcement Learning in a Handful of Trials using Probabilistic Dynamics Models." Advances in Neural Information Processing Systems 31 (2018).
>
> 3. Morgan, Andrew S., Daljeet Nandha, Georgia Chalvatzaki, Carlo D'Eramo, Aaron M. Dollar, and Jan Peters. "Model Predictive Actor-Critic: Accelerating Robot Skill Acquisition with Deep Reinforcement Learning." arXiv preprint arXiv:2103.13842 (2021).
>
> 4. Lambert, Alexander, Adam Fishman, Dieter Fox, Byron Boots, and Fabio Ramos. "Stein variational model predictive control." arXiv preprint arXiv:2011.07641 (2020).

---

> > ### Comment · Reviewer_xm2M · 2021-09-03
> > **Response to Rebuttal**
> >
> > Thank you for the detailed response. This looks fine to me. I will keep my score.

---

### Author Response · Authors · 2021-08-24
**Further Experimental Details**

We performed extra experiments and ablation studies based on reviewer feedback. The results can be found at the following specific links:

**All experiments**: https://sites.google.com/view/manipulation-mpc/further-experimental-results

**Dynamic Obstacle**: https://sites.google.com/view/manipulation-mpc/further-experimental-results#h.1ftgxyyxsj65

**Cost Ablations**: https://sites.google.com/view/manipulation-mpc/further-experimental-results#h.ud1wkx4676yl

**Number of Particles**: https://sites.google.com/view/manipulation-mpc/further-experimental-results#h.js1nubk6xyr8

**Horizon length**:  https://sites.google.com/view/manipulation-mpc/further-experimental-results#h.ugmxgp7scqxs

**Sampling Strategy**: https://sites.google.com/view/manipulation-mpc/further-experimental-results#h.cedxyoeznabo

**Compute Timing Benchmark**: https://sites.google.com/view/manipulation-mpc/further-experimental-results#h.3v3jt4qbc7zu

---

### Author Response · Authors · 2021-08-31
**Revision uploaded**

We thank all the reviewers and the metareviewer for their thoughtful comments on our work. **We have uploaded a revision to our paper.** In summary, we have made the following changes to the supplementary material:

1. **Ablation Studies:** We have added the results of our extensive ablation studies to Section 5 of the supplementary material. The results can also be found on our website following the links in our comments below.

2. **Related Work:** We have added the relevant related work mentioned by Reviewer 4 (U2Zx), specifically Pinneri et. al, 2020 in Section 3.

3. **Ball Dynamics:** Based on the comments from Reviewers 2 (poXv) and 4(U2Zx) we have provided a more detailed explanation of the ball dynamics model used for the dynamic balancing task in Section 4.1.

4. **Collision Avoidance Cost Training:** Based on the comment from Reviewer 3 (sfFw), we provide details on the pre-trained SceneCollisionNet model used for the environment collision avoidance cost in Section 4.3.

---

> ### Comment · Reviewer_sfFw · 2021-09-03
> **Can't see the updated version**
>
> Hi,
>
> Are you sure that you've uploaded the new version? I cannot find Section 4.3 in the pdf.

---

> > ### Author Response · Authors · 2021-09-03
> > **Section 4.3 in supplementary material**
> >
> > Hi, the section 4.3 is in the supplementary material pdf.

---

### Meta-Review · Area_Chair_YVrg · 2021-08-13

**Recommendation:** Accept (Oral)
**Confidence:** 4

**Metareview:**


The paper provides a new framework for sampling-based model predictive control applied to robot manipulation. To efficiently find a policy over a fixed time horizon in realtime, the authors use a simplified robot model by using forward dynamics and to leverage low-discrepancy sampling to find valid control sequences (policies) over the desired time horizon. Authors proposed different cost terms to achieve better manipulation behaviour increasing the sampling efficiency of sample-based MPC. The presented work includes a set of elements to improve the performance of Sampling-Based MPC methods for robots  with higher DOFs. The paper does not explain how the learned components for collision avoidance are trained. The authors need to clearly explain how their work is different from the prior works mentioned in the reviewers’ comments.



**quality**: Good, but some references to the prior works are missing.

**Clarity**:  Good: Minor revision needed to improve organisation or clarity.

**Significance**:  The integration of different components and real-robot experimentations are interesting, but the theoretical significance is not clear in the current version of the paper.

**Originality**:  The novelty and contribution of the paper are questioned by reviewers. Clarifying this may improve the paper.

**Post reviewer discussion** the response to the reviewer's comments clarified many points.

---

> ### Author Response · Authors · 2021-08-25
> **Response to meta review**
>
> We would like to thank the area chair for providing feedback on our paper. We have addressed the reviewer-specific concerns in the comments.
>
> We would like to emphasize the fact that the primary aim of our work is to develop a control framework that is scalable to real-time reactive control for manipulators. We demonstrate this through a series of hard and dynamic real-world tasks where our framework is the first to successfully apply sampling based control in the full joint space of a manipulator. While we provide a particular instantiation of the framework in the paper with effective design choices to be leveraged by practitioners, it is extensible beyond that to incorporate different control algorithms, models, and cost functions as well as learned components while maintaining low control latency which is critical for real world robots. Furthermore, our open-source codebase provides a flexible toolkit to researchers to easily start applying such algorithms to their manipulators, and provides full functionality for reactive motion generation, obstacle avoidance and extending to different control algorithms. We believe that such systems implementations are invaluable for advancing real-world robotics research.
>
> Based on the feedback from the reviewers, we have also provided extensive ablation studies of the various components of our system such as the different cost terms, control parameters and sampling strategy. We also provide an analysis of the computational gains from our implementation and a demonstration of dynamic obstacle avoidance.

---

### Decision · Program_Chairs · 2021-09-13

**Decision:**

Accept (Oral)

**Comment:**


The paper provides a new framework for sampling-based model predictive control applied to robot manipulation. To efficiently find a policy over a fixed time horizon in realtime, the authors use a simplified robot model by using forward dynamics and to leverage low-discrepancy sampling to find valid control sequences (policies) over the desired time horizon. Authors proposed different cost terms to achieve better manipulation behaviour increasing the sampling efficiency of sample-based MPC. The presented work includes a set of elements to improve the performance of Sampling-Based MPC methods for robots  with higher DOFs. The paper does not explain how the learned components for collision avoidance are trained. The authors need to clearly explain how their work is different from the prior works mentioned in the reviewers’ comments.



**quality**: Good, but some references to the prior works are missing.

**Clarity**:  Good: Minor revision needed to improve organisation or clarity.

**Significance**:  The integration of different components and real-robot experimentations are interesting, but the theoretical significance is not clear in the current version of the paper.

**Originality**:  The novelty and contribution of the paper are questioned by reviewers. Clarifying this may improve the paper.

**Post reviewer discussion** the response to the reviewer's comments clarified many points.